# Remember, Retrieve and Generate: Understanding Infinite Visual Concepts as Your Personalized Assistant

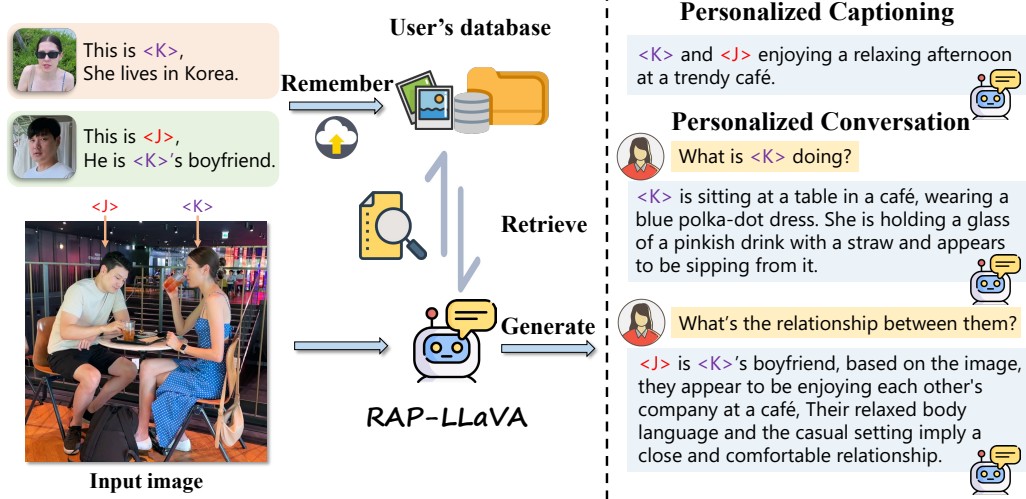

Figure 1: Introduce some user-specific concepts to our RAP-LLaVA, it can remember them and achieve excellent performance in a variety of personalized multimodal generation tasks.

## Abstract

The development of large language models (LLMs) has significantly enhanced the capabilities of multimodal LLMs (MLLMs) as general assistants. However, lack of user-specific knowledge still restricts their application in human's daily life. In this paper, we introduce the **R**etrieval **A**ugmented **P**ersonalization (RAP) framework for MLLMs' personalization. Starting from a general MLLM, we turn it into a personalized assistant in three steps. (a) Remember: We design a key-value database to store user-related information, *e.g.*, user's name, avatar and other attributes. (b) Retrieve: When the user initiates a conversation, RAP will retrieve relevant information from the database using a multimodal retriever. (c) Generate: The input query and retrieved concepts' information are fed into MLLMs to generate personalized, knowledge-augmented responses. Unlike previous methods, RAP allows real-time concept editing via updating the external database. To further improve generation quality and alignment with user-specific information, we design a pipeline for data collection and create a specialized dataset for personalized training of MLLMs. Based on the dataset, we train a series of MLLMs as personalized multimodal assistants. By pretraining on large-scale dataset, RAP-MLLMs can generalize to infinite visual concepts without additional finetuning. Our models demonstrate outstanding flexibility and generation quality across a variety of tasks, such as personalized image captioning, question answering and visual recognition. The code, data and models will be available.

## 1 INTRODUCTION

Recently, the development of large language models (LLMs) has significantly enhanced their language processing and generating capabilities (Zhao et al., 2023b). Building on this foundation, the integration of visual and textual ability through vision-language alignment brings powerful multi-modal LLMs (MLLMs) (Yin et al., 2023; OpenAI, 2023; Gemini-Team, 2024; Liu et al., 2023b; Zhang et al., 2024; Han et al., 2024). MLLMs have shown significant improvement in various tasks, such as image description and question answering, highlighting their potential as human's assistants. However, their lack of user-specific knowledge continues to limit their effectiveness as personalized assistants in daily life.

A qualified personalized assistant first needs to be able to recognize and remember user-related concepts, such as the dog named ⟨Lala⟩ adopted by the user. Although existing MLLMs have been trained on large-scale datasets and possess strong recognition and classification capabilities, directly transferring this knowledge to a user's personal concepts remains challenging. For instance, current leading MLLMs cannot remember your dog's name, even if you have mentioned it before, and they lack awareness of your identity and preferences. Furthermore, the assistant should generate responses tailored to the user's preferences and requirements. However, collecting extensive personal information to train a unique assistant for each user is impractical.

To address this issue, the personalization of MLLMs has become a topic of growing interest, with several approaches already being proposed. MyVLM (Alaluf et al., 2024) utilizes external classification heads to recognize specific concepts, and learns an embedding for each concept to personalize the outputs of vision language models (VLMs). Another concurrent work, Yo'LLaVA (Nguyen et al., 2024), learns a few special tokens to represent each concept. However, both approaches necessitate continuous learning and updating of the model as new concepts emerge. This presents a challenge in dynamic, ever-changing real-world scenarios, where the computing power of users' personal devices is often limited, and all data must be stored locally for privacy concerns.

To address these challenges, we propose the **R**etrieval **A**ugmented **P**ersonalization (RAP), designed to allow MLLMs to update their supported concepts without additional training. Specifically, our RAP works in three key steps. (a) Remember: RAP includes a designed database to help remember each concept via storing its image and basic information, *e.g.*, name, avatar and other attributes. (b) Retrieve: When a user initiates a conversation, RAP will retrieve relevant information from the database using a multimodal retriever. (c) Generate: The input query and retrieved concepts information are incorporated into the MLLM's input for personalized, knowledge-augmented generation. RAP requires only one image per concept with its basic information for personalization. It allows users to make real-time adjustments to the model's outputs by modifying their personal databases, eliminating the need for retraining. A more detailed comparison is presented in Table 1.

Another significant challenge is the lack of large-scale datasets for training MLLMs' personalized generation capabilities. To address this, we design a pipeline to collect extensive training data and create a comprehensive dataset, which enables to train MLLMs to effectively understand and utilize user-related information for generation. Based on this dataset, we train LLaVA (Liu et al., 2023b) and Phi3-V (Rasheed et al., 2024) as novel personalized assistants and evaluate their performance across various tasks, including personalized image captioning, question answering, and visual recognition. Experimental results demonstrate that our RAP-MLLMs excel in wide range of personalized generation tasks, showcasing excellent generation quality and flexibility.

Our contributions are summarized as follows:

- We propose the RAP framework for MLLMs' personalization, allowing models to be trained just once and adapt to diverse users and infinite new concepts without further training.

- We develop a pipeline for collecting large-scale data and create a dataset specifically designed for the personalized training and evaluation of MLLMs. This dataset enables us to train a series of MLLMs to function as personalized assistants.

- Our models demonstrate exceptional performance across various personalized multimodal generation tasks, including personalized image captioning and question answering. Additionally, they exhibit a strong capability to recognize personal concepts within images.

Table 1: **Comparison of Different Personalization Methods.** RAP needs only 1 image with its personalized description, showing outstanding convenience and flexibility in practical applications.

| Method | Number of image | | Data requirement | | | Support |
| | Positive | Negative | Caption | Description | Question-Answer | Real-time edit |
| --- | --- | --- | --- | --- | --- | --- |
| Fine-tuning | n | - | Yes | Yes | No | ✗ |
| MyVLM | n | 150 | Yes | No | Yes | ✗ |
| Yo'LLaVA | n | 200 | No | No | Yes | ✗ |
| RAP(Ours) | 1 | - | No | Yes | No | ✓ |

## 2 RELATED WORK

**Multimodal Large Language Models.** Recently, numerous advanced large language models (LLMs) (Touvron et al., 2023; Zhang et al., 2023b; Chiang et al., 2023; Taori et al., 2023) have been proposed, showing remarkable performance in addressing a wide range of tasks. The rapid development of these LLMs has led to the emergence of multimodal LLMs (MLLMs) (OpenAI, 2023; Gemini-Team, 2024; Liu et al., 2023b; Zhang et al., 2024; Han et al., 2024; Zhu et al., 2023), which excel in general visual understanding and complex reasoning tasks. For instance, LLaVA (Liu et al., 2023b;a) and MiniGPT-4 (Zhu et al., 2023) align visual and language modalities through visual instruction tuning, showcasing impressive capabilities in multimodal conversations. GPT4RoI (Zhang et al., 2023c) and RegionGPT (Guo et al., 2024) enhance fine-grained understanding and reasoning for specific regions by training on region-level instruction datasets. Despite these advancements in tasks such as image captioning and question answering, the lack of user-specific knowledge restricts the generation of personalized content, which hinders the practical application of MLLMs in daily life. In this work, we focus on the personalization of MLLMs, enabling them to remember and understand user-specific concepts, and generate personalized content tailored to user's preferences.

**Personalization of MLLMs.** In the realm of artificial intelligence, personalization typically refers to the process of tailoring a system, application, or model to meet the individual needs and preferences (Yeh et al., 2023; Woźniak et al., 2024; Shi et al., 2024). Substantial efforts have been made to generate images of user's personal objects or in certain context (Ruiz et al., 2023; Kumari et al., 2023; Ham et al., 2024; Gal et al., 2022; Ye et al., 2023). For example, Dreambooth (Ruiz et al., 2023) employs transfer learning in text-to-image diffusion models via fine-tuning all parameters for new concepts. In this paper, we mainly aim at enabling MLLMs to remember and understand user-specific concepts, and generate personalized language outputs. There are several works focusing on the personalization of MLLMs, among which the most relevant works are MyVLM (Alaluf et al., 2024) and Yo'LLaVA (Nguyen et al., 2024). MyVLM introduces the task of personalizing VLMs. It utilizes external classification heads to recognize specific concepts, and learns an embedding for each concept to personalize the outputs of VLMs. Yo'LLaVA personalizes LLaVA by extending its vocabulary and learning specific tokens for each concept. However, both approaches require continuous model updates as new concepts emerge, which presents challenges in dynamic real-world applications. In this work, we propose RAP framework for the personalization of MLLMs, enabling models to be trained once while continuously updating supported concepts without further training.

**Retrieval Augmented Generation.** Retrieval-based methods for incorporating external knowledge have proven effective in enhancing generation across a variety of knowledge-intensive tasks (Gao et al., 2023; Zhao et al., 2023a; Asai et al., 2023; Xu et al., 2023; Yoran et al., 2023; Lin et al., 2023b). DPR (Karpukhin et al., 2020) introduces Dense Passage Retrieval, marking a shift from sparse to dense retrieval techniques. Later, MuRAG (Chen et al., 2022) proposes to use multimodal knowledge to augment language generation. Self-Rag (Asai et al., 2023) introduces special tokens to make retrieval adaptive and controllable. ERAGent (Shi et al., 2024) presents a comprehensive system for retrieval-augmented language models. With the advancements in MLLMs, RAG has been widely applied to multimodal generative tasks. For instance, FLMR (Lin et al., 2023a) employs multi-dimensional embeddings to capture finer-grained relevance between queries and documents, achieving significant improvement on the RA-VQA setting. While existing methods primarily enhance models' performance by retrieving from external knowledge bases, few of them consider the personalization task. Although RAG has been applied to image generation (Blattmann et al., 2022;

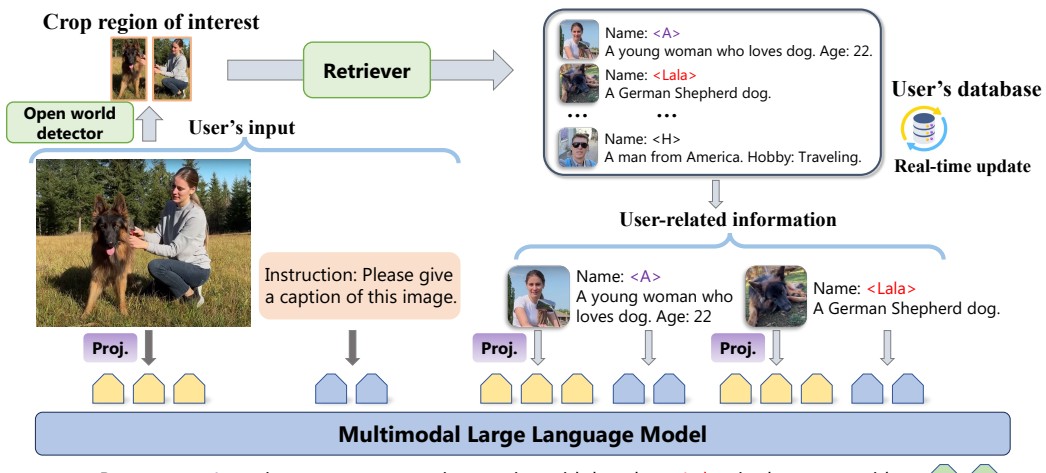

Figure 2: **Retrieval-Augmented Personalization Framework**. Region-of-interest detected by an open world detector are used to retrieve concepts from the database. The images and accompanying information of the retrieved concepts are then integrated into the input for the MLLM.

Zhang et al., 2023a) and image captioning (Li et al., 2024; Ramos et al., 2023), there is currently no existing work focusing on personalizing MLLMs via RAG, to the best of our knowledge.

# 3 RETRIEVAL AUGMENTED PERSONALIZATION

Existing MLLMs typically align other modalities with language. For instance, LLaVA (Liu et al., 2023b) projects visual tokens into text space, and then generates subsequent tokens using an LLM. While these MLLMs perform well in various tasks, the lack of memory and comprehension of personal concepts hinders effective user-specific responses. In this work, we mainly focus on personalizing MLLMs to generate tailored language responses, such as creating personalized captions for user's images and answering questions about personal concepts. In this section, we detail the implementation steps of our proposed Retrieval Augmented Personalization (RAP). Unlike previous approaches that usually necessitate additional data collection and further training to learn new concepts, our RAP does not require additional training as the user's database expands. By pretraining on our dataset, our RAP-MLLMs can adapt to diverse users and infinite new concepts without further training. In section 3.1, we present the RAP framework that is applicable to various types of MLLMs, and then in section 3.2, we provide details of the proposed dataset.

## 3.1 RAP FRAMEWORK

Our RAP works in three main steps: Remember, Retrieve and Generate, as shown in Figure 2.

**Remember.** The premise of personalization is that the model can remember personal concepts and relevant information, such as the dog named ⟨Lala⟩ adopted by ⟨A⟩. To facilitate this, we construct a database $\mathcal{M}$ to store these personal concepts, which comprises an avatar, a name, and a brief description for each concept. The key for each concept in the database is its visual feature, obtained by feeding its image into a pre-trained image encoder $\mathcal{E}(\cdot)$. Examples of our database are presented in Figure 2. When a user initiates a conversation, the input can be represented as $Q = (I, T)$, which may include both image $I$ and some textual instructions $T$. The first step involves identifying possible concepts within the input image that have been previously stored in the database. Previous methods (Alaluf et al., 2024) typically need to learn an external classifier to determine whether a concept appears in the input image, which requires a substantial amount of training data and can only apply to specific concept. To enhance the generalizability of the recognition process, we do not construct specific modules for each concept. Instead, we employ a universal detection model, such as YOLO (Redmon et al., 2016) and YOLO-World Cheng et al. (2024), as recognition model $\mathcal{R}(\cdot)$.

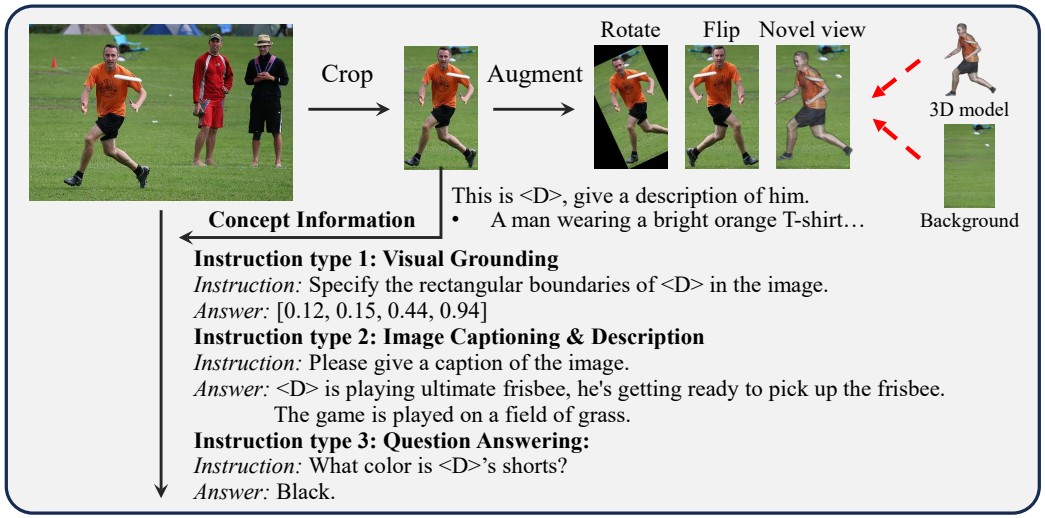

Figure 3: **Our pipeline for data collection.** We first crop the target concept from the image based on the dataset annotations and then query Gemini to generate its personalized description. We also apply data augmentation to diversify these cropped images. Then we combine them with the original image to derive a series of instructions and answers from Gemini.

Given the predefined setting $P$ that specifies which categories should be remembered, the user's region-of-interest can be acquired via $I_u = \mathcal{R}(I, T|P)$.

**Retrieve.** Identified region-of-interest will be used as query to retrieve from the database. For each recognized component $I_u^i$, we feed the image crop into the image encoder $\mathcal{E}(\cdot)$ to get its visual feature $Q^i = \mathcal{E}(I_u^i)$, which is a n-dimensional vector. Then we calculate the euclidean distance between the visual feature and each key $k_j \in \mathcal{M}$, which is calculated as $Dist(Q^i, k_j) = \left\| Q^i - k_j \right\|$. The Top-K image-text pairs $\{(I_1, T_1), (I_2, T_2), \cdots (I_k, T_k)\}$ with the lowest distances are selected. We also introduce retrieval using concept names, such as $\langle sks \rangle$ for a unique concept. When the user mentions the name of an object documented in the database, our model retrieves its related information from the database. This also enables our model to respond to text-only queries effectively.

**Generate.** Each pair $M_j = (I_j, T_j)$ provides related information about a user's personal concept and will be incorporated into the input of the MLLM. Take LLaVA (Liu et al., 2023b) as an example, the image $I_j$ is first encoded by a pre-trained vision encoder, such as CLIP (Radford et al., 2021), to obtain their visual tokens $Z_j$. These image tokens are then projected by a projector into language tokens $H_j^v$, which could be understood by the language model. Simultaneously, corresponding text information $T_j$ are transformed into text tokens $H_j^q$. During training, we keep parameters of the detector and retriever frozen, just train the MLLM. Given the target output sequence $X_a$ of length $L$, the probability of the target answers $X_a$ computed as:

$$p(X_a|I, T, M_1, \cdots M_k) = \prod_{i=1}^{L} p_\theta(X_{a,i}|I, T_{<i}, M_1, \cdots M_k, X_{a,<i})$$

### 3.2 PERSONALIZATION DATASET

Most existing MLLMs struggle to generate personalized outputs even if additional concept information is provided, and there is currently no large-scale dataset for personalized training of MLLMs. To this end, we design a pipeline for data creation and curate a novel dataset specifically for the personalized training and evaluation of MLLMs. We use Gemini-1.5 (Gemini-Team, 2024) to generate annotations for our dataset. An overview of our pipeline and dataset is presented in Figure 3.

The first component of our dataset is dedicated to visual grounding. In this task, a MLLM is trained to determine whether a specific concept is in an image, particularly identifying if the person or object in a reference image appears in the given image. When a positive match is detected, we also require the model to provide the bounding box for the identified concept. For single-concept grounding, we

primarily use the RefCOCO dataset (Kazemzadeh et al., 2014). Based on RefCOCO's annotations, we crop target concepts from the images and assign names to them, which serve as references for specific concepts. We then query Gemini to generate concise descriptions about properties of the concepts in these cropped regions, by which we construct a large-scale database including numerous different concepts. The training data pairs images and these descriptions as queries and the corresponding bounding boxes as outputs. However, data generated in this way is insufficient to simulate the complexity of real-world recognition, especially when the target concept in the reference and input image is captured from different perspectives. To address this, we incorporate the ILSVRC2015-VID video object detection dataset (Russakovsky et al., 2015), TAO (Dave et al., 2020) and CustomConcept101 (Kumari et al., 2023) to enrich our dataset. For multi-object grounding, we use the Object365 dataset (Shao et al., 2019) to construct our training data.

The second component of our dataset is designed for instruction following. This section includes training data for tasks such as image captioning, image description and question answering. For the image captioning and description data, we provide cropped images of target concepts, accompanied by their names and related information from the large-scale database, then query Gemini to generate a caption or description that reflects the concepts depicted in the entire image. For question answering, we first design a set of seed questions to serve as examples. These examples are used to prompt the annotator, Gemini, to generate new questions and corresponding answers. This iterative process facilitates the creation of a rich and diverse collection of conversations that MLLMs can learn from. We construct such data using RefCOCO (Kazemzadeh et al., 2014), Object365 (Shao et al., 2019), TAO (Dave et al., 2020) and CustomConcept101 (Kumari et al., 2023) dataset.

To enhance alignment with real-world scenarios, it is essential to collect data featuring the same identity in various environments. Thus, we also include multiple images about the same individual from the CelebA dataset (Liu et al., 2015) and produce question answering data about the individual. To further diversify the dataset, we apply image editing techniques for data augmentation. This includes performing random rotations and flips on the cropped images, as well as generating novel views of the concepts by diffusion models. Specifically, we use Inpaint-Anything (Yu et al., 2023) to separate the foreground from the background, and use Wonder3D (Long et al., 2024) and Sith (Ho et al., 2024) to synthesize novel views of foreground object or person respectively. Finally, we combine these elements to generate images of the target concept from different perspectives.

In the generation step, the MLLM needs to prioritize accurate and contextually relevant information. Considering that retrieval results can be inaccurate, potentially leading to unreasonable answers, we construct negative samples by incorporating noise elements into the additional input while preserving the original output. This approach trains the model's discrimination capability. By exposing the MLLM to both relevant and irrelevant information during training, it learns to discern and filter out noise, enhancing its robustness at inference time. Additionally, we include a subset of the LLaVA-Instruct-665k visual instruction dataset (Liu et al., 2023a) to retain general knowledge from the original MLLM. Further details about our dataset can be found in Appendix D.

## 4 EXPERIMENT

**Implementation Details.** We conduct experiments on LLaVA-1.5-13B (Liu et al., 2023b) and Phi3-V-3.8B (Rasheed et al., 2024), resulting in two personalized MLLMs, RAP-LLaVA and RAP-Phi3-V. We select YOLO-Worldv2 (Cheng et al., 2024) as the detector and construct a multimodal retriever using Facebook AI Similarity Search (FAISS) (Johnson et al., 2021), employing a pre-trained CLIP ViT-L/14-336 (Radford et al., 2021) as the visual encoder. Due to the context length limitation of the backbone language model, for RAP-LLaVA and RAP-Phi3-V, we retrieve the 2 and 3 different concepts with the highest similarity, respectively. More details can be found in Appendix C.

**Training.** In the training phase, we skip the recognition and retrieval procedures, instead perform instruction tuning to train the MLLMs. We adhere to most settings from the original experiment of LLaVA (Liu et al., 2023b), except for using a maximum learning rate of 1e-4 and training for 1 epoch. We employ low-rank adapters (Hu et al., 2022) to reduce the number of trainable parameters, and train our models on 8 A100 GPUs with a valid batch size of 64.

**Evaluation.** We primarily focus on tasks that require both visual and language understanding. Specifically, we address image captioning and question answering in Section 4.1 and 4.2, and com-

Table 2: **Qualitative Comparison on Image Captioning.** Image examples of target concepts are shown in the left and captions generated are shown in the right.

| Image | Caption |
|---|---|
| 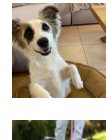 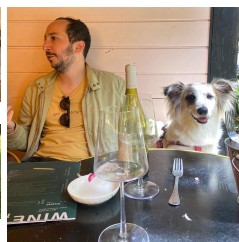 **dog\*** | LLaVA: A man is sitting at a table with a dog, and there are wine glasses and a fork on the table. LLaVA-LoRA: ⟨collie dog⟩ looking pleased as she shares a meal with her owner. MyVLM: ⟨dog*⟩ positioned on a chair by a black table, holding a wine glass in her hand. A white dog sits on the floor beside her... **RAP-LLaVA**(Ours): ⟨dog*⟩ is a very good boy, and he loves to sit at a table with his owner. They are enjoying a meal. |
| 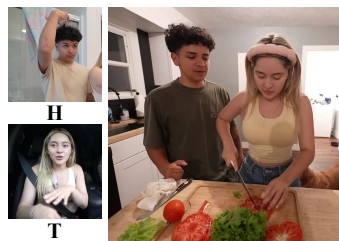 **H** **T** | LLaVA: A man and a woman are standing in a kitchen, preparing food together. The woman is cutting lettuce on a cutting board, while the man watches her. There are several tomatoes ... LLaVA-LoRA: ⟨H⟩ and ⟨K⟩ are preparing a meal together. MyVLM: ⟨T⟩ and her friend ⟨H⟩ are looking very serious as they take in the scenery. **RAP-LLaVA**(Ours): ⟨H⟩ is helping ⟨T⟩ prepare a salad in the kitchen. |
| 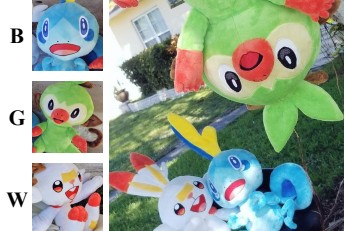 **B** **G** **W** | Phi3-V: A group of stuffed animals, including a blue one, are sitting on a black surface. LLaVA-LoRA: ⟨B⟩, ⟨G⟩ and ⟨W⟩ are happily exploring the grassland. MyVLM: ⟨G⟩ and his crew are always ready to jump into a new adventure. **RAP-Phi3-V**(Ours): ⟨W⟩ is hanging out with ⟨G⟩ and ⟨B⟩ on the lawn. They are having a great time playing! |

pare our models with baseline methods on visual recognition. In Section 4.3, we compare the cost of personalization with existing methods, and present results of ablation studies in Section 4.4.

## 4.1 PERSONALIZED IMAGE CAPTIONING

In this section, we evaluate our models on generating personalized image captions with user's specific concepts. We extend the dataset introduced by MyVLM (Alaluf et al., 2024) via adding 16 new concepts, which include both objects and humans, forming 8 concept pairs that appear together in images. For each pair, there are 8-13 images used for testing. This multiple concepts setting presents additional challenges for personalization.

**Settings.** We compare our models with MyVLM and finetuning based method LLaVA-LoRA (Hu et al., 2022). We do not include Yo'LLaVA since it does not porvide open-sourced model. For LLaVA-LoRA and MyVLM, the training dataset contains 1 image accompanied by 5 captions for each concept. This simulates the real-world challenge of collecting high-quality training data for each concept, which is both difficult and time-consuming. For LLaVA-LoRA, we train it with captions of the training images for 3 epochs, applying low-rank adapters (Hu et al., 2022) and the same hyperparameters as our models. For MyVLM, following their training process, we first train the classification head with the positive and 150 negative images, then train the corresponding concept embedding with the provided captions for each concept. For our models, we construct a database where each concept is represented by a cropped image and a personalized description. Details of our database could be found in Appendix G. All remaining images are used as test samples. This evaluation process is repeated three times using different seeds, and we report the average results.

**Qualitative Comparison.** In Table 2, we present image captions generated by different methods to make a comparison. While LLaVA and Phi3-V generally provides brief and clear captions for most test images, its lack of understanding of the user's specific concepts restricts it from generat-

Table 3: **Quantitative Evaluation on Image Captioning.** We report Recall, Precision and F1-score in the table, the best result in each metric is bold and the second is underlined.

| Method | LLM | Recall | Precision | F1-score |
|---|---|---|---|---|
| LLaVA-LoRA | Vicuna-13B | 82.97 | 93.28 | 87.82 |
| MyVLM | Vicuna-13B | 84.65 | 86.37 | 85.50 |
| RAP-LLaVA | Vicuna-13B | **93.51** | **96.47** | **94.97** |
| RAP-Phi3-V | Phi3-V-3.8B | 88.14 | 95.10 | 91.49 |

Figure 4: Performance under **varying number of personalized concepts.**

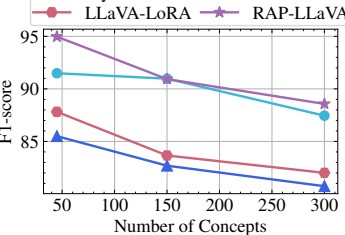

Table 4: **Quantitative Evaluation on Question Answering and Visual Recognition.** The best result in each setting is bold and the second is underlined. Evaluation results of GPT-4V are also provided as reference. Weighted results are computed as arithmetic means.

| Method | Train | #Image | Question Answering | | | Visual Recognition | | |
|---|---|---|---|---|---|---|---|---|
| | | | Visual | Text | Weighted | Positive | Negative | Weighted |
| GPT-4V+Prompt | ✗ | 1 | 0.866 | 0.982 | 0.924 | 0.809 | 0.992 | 0.901 |
| GPT-4V+Prompt | ✗ | 5 | 0.887 | 0.987 | 0.937 | 0.851 | 0.998 | 0.925 |
| LLaVA | ✗ | - | 0.899 | 0.659 | 0.779 | 0.000 | **1.000** | 0.500 |
| LLaVA-LoRA | ✓ | 1 | 0.900 | 0.583 | 0.741 | 0.988 | 0.662 | 0.825 |
| LLaVA-LoRA | ✓ | 5 | 0.935 | 0.615 | 0.775 | **0.997** | 0.444 | 0.721 |
| MyVLM-LLaVA | ✓ | 5 | 0.912 | - | - | 0.994 | 0.845 | 0.919 |
| Yo'LLaVA | ✓ | 5 | 0.929 | 0.883 | 0.906 | 0.949 | 0.898 | 0.924 |
| RAP-LLaVA(Ours) | ✗ | 1 | 0.935 | **0.938** | **0.936** | 0.979 | 0.982 | **0.980** |
| RAP-Phi3-V(Ours) | ✗ | 1 | **0.941** | 0.850 | 0.896 | 0.922 | 0.988 | 0.955 |

ing a more personalized caption. LLaVA-LoRA and MyVLM can generate personalized captions, however, the limited training data often results in imprecise outputs, particularly noticeable when multiple concepts are present in the same image. In contrast, our models produce clear and accurate captions based on the database content, which also ensures the reliability of the outputs. Additional examples of personalized captions generated by the models could be found in Appendix E.

**Quantitative Evaluation.** We employ recall, precision and the comprehensive metric F1-score as our evaluation metrics. Recall is calculated as the percentage of correct occurrences of target concepts, while precision is the ratio of correct concept names to the total number of concept names presented. The experimental results are shown in Table 3. From the results, we find that the finetuning based model LLaVA-LoRA achieves higher performances than MyVLM. Notably, the classification heads of MyVLM exhibit higher error rates when the number of positive images is limited, leading to weaker performance. Our models demonstrate superior performance in both recall and precision metrics, highlighting the advantages of our RAP-MLLMs in data efficiency.

**Influence of Number of Learned Concepts.** In real-world scenario, users' personal databases typically expand over time. Next, we evaluate the performance of various methods with varying numbers of learned concepts. We extend the database with hundreds of new concepts selected from RefCOCO dataset (Kazemzadeh et al., 2014), ensuring no overlap with the test dataset. For LLaVA-LoRA and MyVLM, we provide images containing the target concepts along with their captions as training data, and we assess the models' performance on the original test dataset. The results are presented in Figure 4. As the number of learned concepts increases, performance of all methods declines. More learned concepts result in increased recognition errors, leading to a drop in performance. Our RAP-MLLMs maintain the highest performance under different settings.

### 4.2 PERSONALIZED QUESTION ANSWERING

**Settings.** In this section, we evaluate different methods on the benchmark of personalized question answering introduced by Yo'LLaVA (Nguyen et al., 2024), which contains both visual-based and text-only questions about user's personal concepts. For each concept, we generate a description that

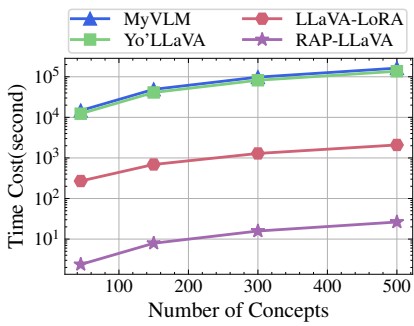
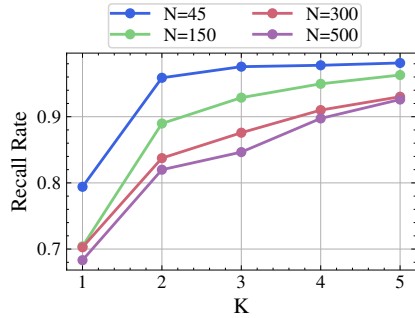

Figure 5: **Time Cost of Personalization**. We conduct experiment with 2 A800 GPUs.

Figure 6: **Performance of Our Retriever**. Top-K recall rates under varying database size N.

serves as the concept's information in our database. For LLaVA-LoRA, we feed these descriptions and corresponding images to train the model to describe the properties of concepts. Additionally, we incorporate text-only queries and answers to enhance the model's understanding of specific concepts from textual perspectives. The training dataset for Yo'LLaVA and MyVLM consists of 5 positive images with question answering pairs and 200 negative images for each concept. For GPT-4V (OpenAI, 2023), images and related information about the concepts mentioned in the questions are provided as supplementary prompt. Additional details on the baselines are provided in Appendix C.

**Results and Analysis.** The experimental results are provided in Table 4. LLaVA and LLaVA-LoRA both perform well in visual based question answering, because substantial information of the target concept can be obtained from the images. However, their performance is quite poor when images of the target concept mentioned in the question are not available. MyVLM performs well in visual question answering but does not support text-only question answering. Yo'LLaVA excels in text-only question answering, but its performance is still limited by the insufficient information provided by the learned tokens of a concept. In contrast, our models demonstrate balanced performance in both visual and text-only question answering. By providing a single image, our RAP-LLaVA surpasses baseline methods and achieves performance comparable to that of GPT-4V.

**Visual Recognition.** We also evaluate the models' recognition abilities for a more comprehensive comparison. In this task, the MLLMs are required to determine whether a personal concept exists in an image. We query them with "Is ⟨sks⟩ in the image? Answer with a single word or phrase.", where ⟨sks⟩ is replaced by corresponding concept name. For positive images, the desired response is "Yes" and "No" for negative. Results show that, without understanding of personal concepts, the vanilla LLaVA consistently outputs negative responses. After training on the target concepts, LLaVA-LoRA, MyVLM and YoLLaVA tend to give positive responses, but struggle to differentiate between concepts, resulting in weaker performance on negative images. Our models demonstrate exceptional performance in both positive and negative scenarios, achieving the best overall results.

### 4.3 COST OF PERSONALIZATION.

We further compare the costs of personalization. As shown in table 1, existing methods usually struggle with continuous updates or have high demands for training data. For finetune-based method like LLaVA-LoRA, while they can achieve satisfactory performance, finetuning the model each time a new concept emerges incurs substantial computational costs. MyVLM and Yo'LLaVA learn an embedding or some new tokens to represent the new concept without updating the pre-trained MLLM's parameters, however, they require multiple labeled images of the target concept and a large number of negative images, which poses significant challenges for data collection. In contrast, our RAP requires only 1 image with its related information provided by the user, achieving outstanding performance across various personalized generation tasks. At the same time, by modifying images and descriptions in the database, RAP enables real-time editing of personalized generation settings. We present examples of real-time concept editing in Table 10.

**Time Cost.** We also evaluate the time cost associated with different methods for learning a set of user's concepts. The results are presented in Figure 5. MyVLM has to train an external recognition model for each concept and learn an embedding to adjust the model's outputs. Similarly, Yo'LLaVA

Table 5: We evaluate model's performance with perfect retrieval, and test contributions of each dataset component.

| Setting | Recall | Precision | F1-score |
|---|---|---|---|
| RAP-LLaVA | 93.51 | 96.47 | 94.97 |
| *Skip retrieval* | 96.16 (+2.7) | 100.0 (+3.5) | 98.04 (+3.1) |
| *- Data aug* | 89.25 (-4.3) | 98.01 (+1.5) | 93.42 (-1.6) |
| *- Neg samples* | 95.74 (+2.2) | 58.21 (-38.3) | 72.40 (-22.6) |

Table 6: **Evaluation on Multimodal Benchmarks**. RAP-LLaVA maintains most knowledge of original LLaVA.

| Method | MMMU | InfoSeek |
|---|---|---|
| LLaVA | 0.364 | 0.205 |
| LLaVA-LoRA | 0.359 | 0.205 |
| RAP-LLaVA | 0.361 | 0.218 |
| RAP-LLaVA(With KB) | **0.369** | **0.344** |

needs to learn new tokens for each concept. During the optimization process, both approaches necessitate multiple forward and backward pass of the MLLM, resulting in significant time consumption. In contrast, our RAP only requires time for encoding the image and adding its embedding to the database, which can be accomplished in just a few seconds. This significantly enhances the convenience and practicality of our models in practical applications.

## 4.4 ABLATION STUDY.

**Retriever.** The recall rate of the retriever is crucial for a RAG system. We first assess the retriever's performance on the personalized captioning dataset. We use the detection model to identify potential concepts and retrieve the K concepts with the highest similarity from the database. The Top-K recall rates for varying values of K and database sizes N are presented in Figure 6. Results indicate that as the database size increases, the retriever's performance declines, while a larger K generally enhances the recall rate. Notably, even with 500 personal concepts to remember, the Top-5 recall rate is still able to surpass 90%, which guarantees the effectiveness of our RAP framework.

**Generation Ability of MLLM.** We skip the recognition and retrieval processes, providing the MLLM with relevant information of each concept present in the image to evaluate the generation capability of the trained MLLM. The results, shown in Table 5, indicate that when relevant concept information is supplied, our RAP-LLaVA achieves superior generation performance, obtaining 100% precision without outputting irrelevant concepts as well as a higher recall rate.

**Dataset Composition.** We conduct experiments to assess contribution of each component in our dataset. First, we remove data generated through data augmentation and train the original LLaVA. The results indicate a obvious decrease in the recall metric for image captioning, resulting in lower overall performance. We further exclude constructed negative samples from the dataset and retrain the model, then we find that it performs poorly on precision metric. This suggests a diminished ability to discriminate against noisy concepts not present in the image.

**Multimodal Benchmark.** We also evaluate our model's performance on several traditional multimodal benchmarks, including MMMU (Yue et al., 2024) and InfoSeek (Chen et al., 2023). We assess our models' performance both with and without external knowledge base. Details of the knowledge base are provided in Appendix C. We evaluate on the validation set of MMMU, and 5K questions sampled from the validation set of InfoSeek. We use the official scripts to get the results, which are presented in Table 6. From the results, our RAP-LLaVA retains most general knowledge of the original LLaVA. It also equips the MLLM with the ability to retrieve information from an external knowledge base, demonstrating superior performance in knowledge intensive tasks.

## 5 CONCLUSION

In this paper, we introduce the RAP framework for personalizing MLLMs. This framework enables MLLMs to understand an infinite number of user-specific concepts, generate personalized captions and respond to user-related queries. To enhance the quality of the generated content and better align outputs with user's configuration, we curate a large-scale dataset for personalized training of MLLMs. Using this dataset, we train a series of MLLMs to function as personalized assistants. Experimental results show that RAP-MLLMs achieve exceptional performance in various personalized generation tasks while preserving the general knowledge of the original MLLMs. Moreover, our RAP framework allows real-time adjustments to generation settings. It eliminates the need for retraining on new concepts and provides significant flexibility in personalized generation.

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

# A    APPENDIX OVERVIEW

- Section B: Additional evaluations of our models.
- Section C: More experiment details.
- Section D: More details of RAP dataset.
- Section E: Additional demonstrations.
- Section F: Analysis on limitations of our work.
- Section G: Examples of the personalized database.

# B    ADDITIONAL EVALUATION RESULTS

Table 7: Ablation studies on **Question Answering and Visual Recognition.** Weighted results are computed as arithmetic means.

| Method | Question Answering | | | Visual Recognition | | |
|---|---|---|---|---|---|---|
| | Visual | Text | Weighted | Positive | Negative | Weighted |
| RAP-LLaVA | 0.935 | 0.938 | 0.936 | 0.979 | 0.982 | 0.980 |
| *- Data aug* | 0.924 (-0.011) | 0.918 (-0.020) | 0.921 (-0.015) | 0.943 (-0.036) | 0.988 (+0.006) | 0.965 (-0.015) |
| *- Neg samples* | 0.918 (-0.017) | 0.933 (-0.005) | 0.925 (-0.011) | 0.958 (-0.021) | 0.985 (+0.003) | 0.971 (-0.009) |

**Ablation Studies**. We conduct ablation experiments on the question answering and recognition benchmark, experimental results are present in Table 7. The results further demonstrate that our data augmentation and the constructed negative samples also contribute to the model's performance.

# C    MORE EXPERIMENTAL DETAILS

**Implementation details.** We utilize YOLO-Worldv2-X (Cheng et al., 2024) as the detection model, setting detection classes to include all categories stored in the database to reduce the interventions from unrelated objects. We construct a multimodal retriever using Facebook AI Similarity Search (FAISS) (Johnson et al., 2021), employing a pre-trained CLIP ViT-L/14-336 (Radford et al., 2021) as the visual encoder. Each key in the database is generated by inputting the image of a concept into the CLIP visual encoder, resulting in a 768-dimensional vector. Considering the restriction of context length of the backbone language model, we retrieve the 2 most similar images from the database for each region of interest. And then, we select 2 and 3 different concepts with the highest similarity among all as supplementary inputs for RAP-LLaVA and RAP-Phi3-V, respectively.

**External knowledge base.** For MMMU (Yue et al., 2024), we use 30K images paired with corresponding captions from Wikipedia as the external knowledge base. During testing, we retrieve the three most similar images based on the question's image and incorporate only the textual knowledge to the input. For InfoSeek (Chen et al., 2023), we randomly sample 5K questions from the validation set and construct a knowledge base containing 50K entities from Wikipedia database provided by the authors, which includes all relevant entities associated with the questions. For each question, we retrieve the most similar entity and add only the textual knowledge to the input.

**Baselines.** For MyVLM, we find that when the training data is very limited, it is quite hard for the classification head to work effectively. Therefore, we use data augmentation to help improve its performance. Specifically, we crop the single image into several pieces containing the target concept to improve the accuracy of classification heads. To distinguish between multiple possible different concepts that may appear in the image, we use ⟨sks1⟩, ⟨sks2⟩. . . as concept identifiers. For YoLLaVA, as there is no open-source code or model available, we present its experimental results as reported in the original paper (Nguyen et al., 2024).

## D    DETAILS OF DATASET

### D.1    DATASET COMPOSITION

- We provide a summary of the composition of our dataset in Figure 7, which visually represents the distribution of different components.

- Table 8 presents detailed numerical data for each part.

- In Table 9, we specify the sources for each component of our dataset.

Figure 7: Composition of our dataset.

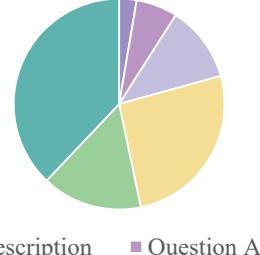

- ■ Image Description  ■ Question Answering
- ■ Image Captioning  ■ LLaVA-Instruction
- ■ Recognition  ■ Visual Grounding

Table 8: Statistics of our dataset.

| Type | Size |
|---|---|
| Visual Grounding | 100K |
| Recognition | 40K |
| Caption & Description | 37K |
| Question Answering | 16K |
| LLaVA-Instruction | 67K |
| **Total** | **260K** |

Table 9: Data source.

| Type | Source Dataset |
|---|---|
| Visual Grounding | RefCOCO (Kazemzadeh et al., 2014), TAO (Dave et al., 2020) ILSVRC2015-VID (Russakovsky et al., 2015), Object365 (Shao et al., 2019) |
| Recognition | CustomConcept101 (Kumari et al., 2023), CelebA (Liu et al., 2015) |
| Caption & Description | RefCOCO (Kazemzadeh et al., 2014), TAO (Dave et al., 2020) Object365 (Shao et al., 2019), CustomConcept101 (Kumari et al., 2023) |
| Question Answering | RefCOCO (Kazemzadeh et al., 2014), TAO (Dave et al., 2020) Object365 (Shao et al., 2019), CustomConcept101 (Kumari et al., 2023) CelebA (Liu et al., 2015) |
| LLaVA-Instruction | LLaVA-Instruct-665K (Liu et al., 2023a) |

### D.2    INSTRUCTIONS

In this section, we present the instruction templates used to create our dataset:

- Table 17 contains instructions for visual grounding and recognition.

- Table 18 includes example instructions for image captioning.

- Table 19 presents example instructions for image description.

- Table 20 presents example questions used for question answering synthesis.

## E    ADDITIONAL DEMONSTRATIONS

In this section, we provide more qualitative results obtained by various models.

- In Table 10, we demonstrate how our models achieve real-time editing of concepts by modifying the database.

- In Table 11, we demonstrate the real-time addition of new concepts by updating the database.

- In Table 13, we present additional image captions generated by RAP-LLaVA and other methods.

Table 10: **Examples of Concept Editing**. Based on the information recorded in the database, our RAP-LLaVA can provide reliable and accurate answers.

| Image | Concept | Information in the Database |
|---|---|---|
| 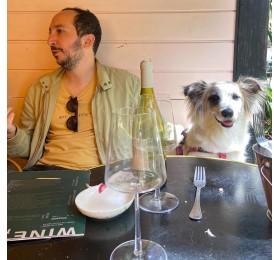 | dog* | 1. A white and gray dog with long fur. He has black eyes. His favorite food is chicken. 2. A white and gray dog with long fur. He has black eyes. His favorite food is beef. |
|  |  | Question: What is ⟨dog*⟩'s favorite food? |
|  |  | 1. **RAP-LLaVA**: Chicken. |
|  |  | 2. **RAP-LLaVA**: His favorite food is beef. |

Table 11: **Examples of Concept Updating**. The first caption is generated when toy2 not yet stored in the database. Once the new concept is added, RAP-LLaVA can recognize both toy1 and toy2.

| Image | Concept | Information in the Database |
|---|---|---|
| 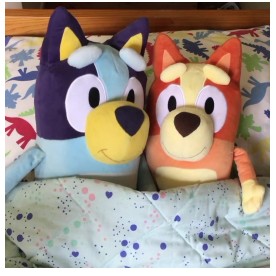 | toy1 | A plush toy. It is orange with a yellow belly and a brown nose. |
|  | toy2 | This is a plush toy of the bluey character. It is a light blue color with a purple patch on its head, and its ears are yellow. |
|  |  | Question: Give a caption of this image. |
|  |  | 1. **RAP-LLaVA**: ⟨toy1⟩ is ready for bed! He's snuggled up with his friend, a blue and yellow dog plushie. They're both looking so cozy and ready for a good night's sleep! |
|  |  | (Add new concept ⟨toy2⟩ to the database) |
|  |  | 2. **RAP-LLaVA**: ⟨toy1⟩ and ⟨toy2⟩ are ready for bed! They are both snuggled under the covers with their favorite stuffed animals. |

- In Table 14, we present additional image captions generated by RAP-Phi3-V and other methods.
- In Table 15, we provide demonstrations of image description generated by RAP-LLaVA and LLaVA.
- In Table 12, we present qualitative results on personalized conversation of our models.
- In Table 16, we provide results on visual recognition of RAP-LLaVA. It also has the ability to give precise bounding box of specific concept in the image.

## F    LIMITATION

Our proposed RAP framework is a retrieval-based method. The limitations of RAP mainly concern the additional computational cost of generation and the precision of the retriever. While incorporating external information effectively generates more specific answers, it inevitably increases the context length for MLLMs, leading to additional computational overhead during the generation pro-

cess. We will further explore ways to mitigate this computational burden. Another limitation is the personalization performance of our RAP-MLLMs depends on the retriever's capability This proposes need for a robust multi-modal retriever that can discern intricate features to enhance retrieval precision. Despite these limitations, RAP offers a timely solution for MLLM personalization. By retrieving from a user's specific database, RAP facilitates reliable and flexible personalized generation, which is valuable in practical applications.

## G EXAMPLES OF THE PERSONALIZED DATABASE

We give some visualized examples of our database in Table 21. For each concept in the database, users need to provide an image with its name and an optional personalized description to give additional information. During inference, the images, names and other information of retrieved concepts are integrated into the input for the MLLM. Users have the flexibility to define the name and personalized description based on their preferences, and our RAP-MLLMs will generate answers according to the provided information.

Table 12: Examples of personalized conversations obtained by RAP-LLaVA.

| Image | Concept | Information in the Database |
|---|---|---|
|  | my cat | A tabby cat with its eyes closed is laying on a light grey surface. The cat's fur is a mix of light and dark brown stripes. |

| Image | Concept | Information in the Database |
|---|---|---|
|  | | Question: Where is ⟨my cat⟩?
**RAP-LLaVA**: ⟨my cat⟩ is on a staircase.

Question: What is the type of ⟨my cat⟩?
**RAP-LLaVA**: ⟨my cat⟩ is a tabby cat.

Question: What is the type of ⟨my cat⟩?
**RAP-LLaVA**: ⟨my cat⟩ is a tabby cat with a mix of light and dark brown stripes. |

| Image | Concept | Information in the Database |
|---|---|---|
|  | A | ⟨A⟩ is a smiling Asian man with short black hair, wearing glasses and a grey sweater. |

| Image | Concept | Information in the Database |
|---|---|---|
|  | | Question: Can you see ⟨A⟩ in this photo?
**RAP-LLaVA**: Yes, ⟨A⟩ is captured in this image.

Question: What color is ⟨A⟩'s shirt?
**RAP-LLaVA**: ⟨A⟩ is wearing a purple shirt.

Question: What is ⟨A⟩'s right hand holding?
**RAP-LLaVA**: Chopsticks. |

| Image | Concept | Information in the Database |
|---|---|---|
|  | dug | ⟨dug⟩ is a golden retriever from the movie Up. He has a brown coat and a big, goofy smile, wearing a collar with some round objects. |
| (Text-only conversation) | | Question: Can you describe ⟨dug⟩ in detail?
**RAP-LLaVA**: ⟨dug⟩ is a golden retriever from the movie Up. He has a brown coat and a big, goofy smile. He is wearing a collar with some round objects. He looks very happy and playful. |

| Image | Concept | Information in the Database |
|---|---|---|
|  | C | ⟨C⟩ is a young woman with long blonde hair, wearing a white sleeveless shirt. She has bright red lipstick and light makeup. |
| (Text-only conversation) | | Question: What color is ⟨C⟩'s hair?
**RAP-Phi3-V**: ⟨C⟩'s hair is blonde.

Question: What color of clothes does she like?
**RAP-Phi3-V**: She likes white clothes. |

Table 13: Additional qualitative comparison on image captioning between RAP-LLaVA and other methods.

| Image | Caption |
|---|---|
| 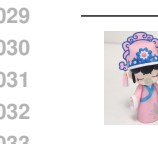 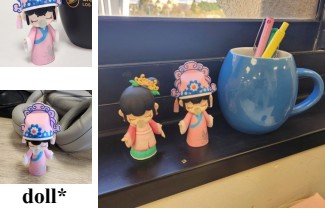 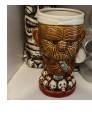 **doll*** | LLaVA: A blue coffee cup sits next to two dolls on a window sill.
LLaVA-LoRA: ⟨doll*⟩ positioned on a windowsill with a blue mug containing pencils
MyVLM: ⟨doll*⟩ is found sitting beside a window ledge with a cup and toothbrushes nearby. In the image, ⟨doll*⟩ appears to be peacefully resting, possibly enjoying the view or taking a break.
**RAP-LLaVA**(Ours): ⟨doll*⟩ and her friend are ready for a fun day at the office! They're ready to take on any challenge that comes their way. |
| 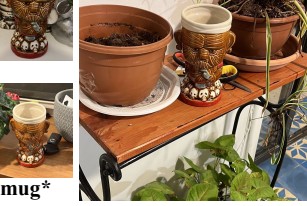 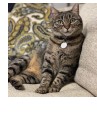 **mug*** | LLaVA: A wooden shelf holds several potted plants, including a large clay pot and a small vase. The shelf is positioned near a wall, and the plants are arranged in a visually appealing manner.
LLaVA-LoRA: ⟨mug*⟩ on a wooden shelf next to a plant and two potted plants.
MyVLM: ⟨mug*⟩ on a a blue and white tiled floor next to indoor plants, a shelf with pots and a vase.
**RAP-LLaVA**(Ours): A brown ceramic tiki mug with the face of a tiki head, ⟨mug*⟩, sits on a wooden shelf with two potted plants. |
| 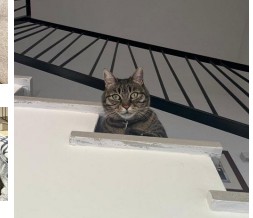 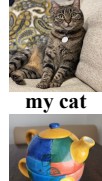 **my cat** | LLaVA: A cat is sitting on a ledge near a staircase.
LLaVA-LoRA: ⟨my cat⟩ sitting on a staircase, looking down.
MyVLM: ⟨my cat⟩ in a typical pose, gripping the banister of a staircase. He is wearing a collar.
**RAP-LLaVA**(Ours): ⟨my cat⟩ is relaxing on the stairs. Look at those eyes! What a beautiful cat. |
| 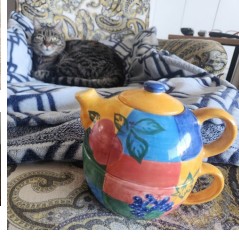 **my cat** 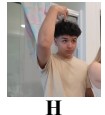 **teapot*** | LLaVA: A cat is laying on a blanket on a couch, next to a colorful teapot.
LLaVA-LoRA: ⟨my cat⟩ reclining on a chair with a ⟨teapot*⟩ beside
MyVLM: ⟨teapot*⟩ on the couch near an orange, yellow, and blue teapot. The cat is laying on a blanket, and the teapot is placed on a table.
**RAP-LLaVA**(Ours): ⟨my cat⟩ is taking a well-deserved nap next to ⟨teapot*⟩. |
| 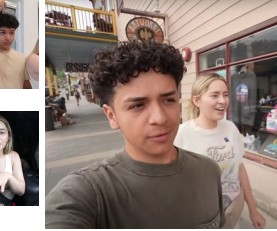 **H** 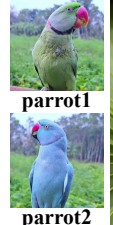 **T** | LLaVA: A man and a woman are walking down a street, with the man taking a selfie using his cell phone. They are both smiling as they walk, and the man is holding his phone . . .
LLaVA-LoRA: ⟨H⟩ and ⟨K⟩ are looking happy as they walk down the street together.
MyVLM: ⟨H⟩ and ⟨T⟩ are looking very serious as they take in the scenery.
**RAP-LLaVA**(Ours): ⟨T⟩ is walking down the street with her friend ⟨H⟩. |
| 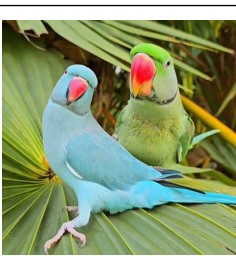 **parrot1** **parrot2** | LLaVA: Two colorful parrots are perched on a palm tree, sitting next to each other.
LLaVA-LoRA: ⟨parrot1⟩ and ⟨parrot2⟩ are looking very serious as they perch on a palm frond.
MyVLM: ⟨parrot1⟩ and ⟨parrot2⟩ look so cute together.
**RAP-LLaVA**(Ours): ⟨parrot1⟩ and ⟨parrot2⟩ are both beautiful birds. ⟨parrot1⟩ has a bright red beak and ⟨parrot2⟩ has a light blue head and pink feet. They are both perched on a branch. |

Table 14: Additional qualitative comparison on image captioning between RAP-Phi3-V and other methods.

| Image | Caption |
|---|---|
| 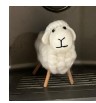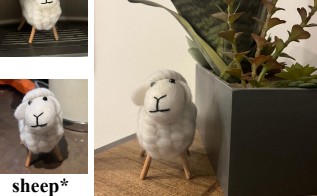**sheep*** | Phi3-V: A small white sheep figurine is sitting on a wooden table next to a plant.
LLaVA-LoRA: ⟨sheep*⟩ positioned next to a potted plant on a wooden table.
MyVLM: ⟨sheep*⟩ positioned within a square shape within a vase made of metal positioned on a table made of wood.
**RAP-Phi3-V**(Ours): ⟨sheep*⟩ is ready to play on the shelf with the plant! |
| 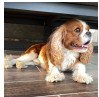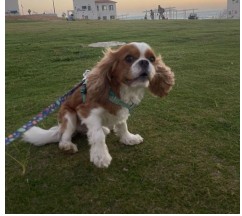**dog1** | Phi3-V: A brown and white dog is sitting on the grass with a leash attached to it.
LLaVA-LoRA: ⟨dog1⟩ sitting on the grass with a look of determination.
MyVLM: on a cheerful note, ⟨dog1⟩ restfully resides, surrounded by the soothing sounds of nature.
**RAP-Phi3-V**(Ours): This is a picture of ⟨dog1⟩, a cute dog with light brown and white fur. He looks happy and playful. |
| 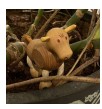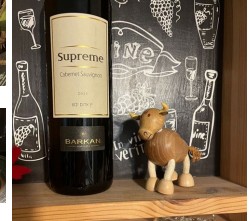**bull*** | Phi3-V: A bottle of wine is displayed next to a wooden cow statue.
LLaVA-LoRA: ⟨bull*⟩ positioned next to a bottle of supreme cabernet sauvignon.
MyVLM: ⟨gold pineapple⟩ pars the bottle of wine. A bottle of wine sits next to a bottle of wine.
**RAP-Phi3-V**(Ours): ⟨bull*⟩ stands guard over a bottle of SUPREME wine. |
| 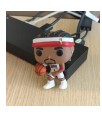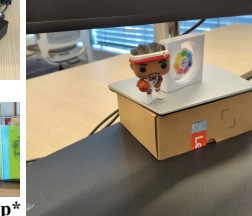**funko pop*** | Phi3-V: A small figurine of a basketball player is placed on top of a box.
LLaVA-LoRA: ⟨funko pop*⟩ sitting on a desk next to a box and a monitor.
MyVLM: ⟨funko pop*⟩ stands out in a photo set against a dark background.
**RAP-Phi3-V**(Ours): ⟨funko pop*⟩ is ready to dominate the court! |
| 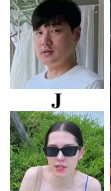**J**
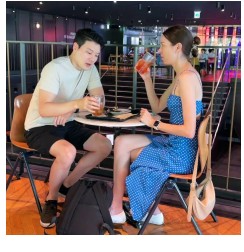**K** | Phi3-V: A man and woman are sitting at a table in a restaurant, having a conversation and enjoying their time together.
LLaVA-LoRA: ⟨J⟩ and ⟨K⟩ enjoying a meal together.
MyVLM: ⟨J⟩ and ⟨K⟩ enjoying their time together.
**RAP-Phi3-V**(Ours): ⟨J⟩ and ⟨K⟩ are enjoying a drink at a bar. |
| 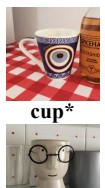**cup***
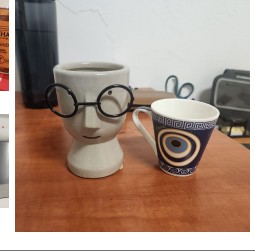**head*** | Phi3-V: A small statue of a man wearing glasses is sitting next to a cup on a table.
LLaVA-LoRA: A table hosts ⟨cup*⟩ and a sculpture of a head wearing glasses.
MyVLM: A table with ⟨cup*⟩, an espresso cup, and a ceramic mug depicting a face wearing glasses sits on a wooden surface.
**RAP-Phi3-V**(Ours): ⟨head*⟩ and ⟨cup*⟩ are ready for their morning coffee. |

Table 15: Qualitative results of personalized image description obtained by RAP-LLaVA.

| Image | Description |
|---|---|
| 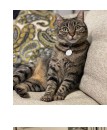 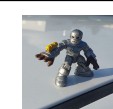 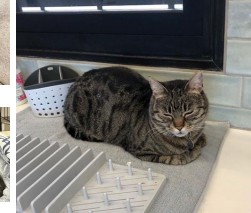 **my cat** | LLaVA: The image features a large, fluffy cat sitting on a counter-top, possibly in a kitchen. The cat is positioned near a sink and appears to be resting or sleeping. The countertop is covered with a towel, providing a comfortable surface for the cat to lay on . . .

**RAP-LLaVA**(Ours): The image shows ⟨my cat⟩, a tabby cat with its eyes closed, laying on a light grey surface. The cat's fur is a mix of light and dark brown stripes. The cat is relaxed and comfortable. |
| 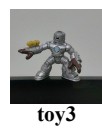 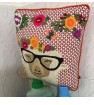 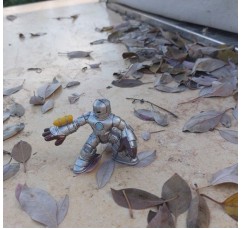 **toy3** | LLaVA: The image features a small metal robot figure, possibly a toy, standing on a table or counter. The robot is positioned in the center of the scene, surrounded by a variety of leaves scattered around it. The leaves are of different sizes and . . .

**RAP-LLaVA**(Ours): The image shows a toy figurine of Iron Man, a superhero from Marvel comics. He is wearing a silver and red suit of armor and is standing on a surface covered in leaves. The toy is called ⟨toy3⟩. |
| 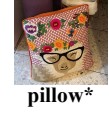 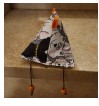 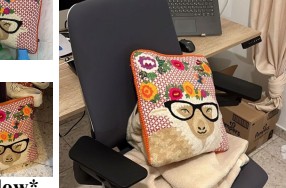 **pillow*** | LLaVA: The image features a chair with a pillow on it, which has a picture of a sheep on it. The chair is positioned in front of a desk with a laptop and a monitor on it. The laptop is placed on the left side of the desk, while the monitor is on the right side . . .

**RAP-LLaVA**(Ours): The image shows a chair with a ⟨pillow*⟩ on it. The ⟨pillow*⟩ is a pattern of flowers, leaves, and a sheep wearing glasses with a flower crown. There is also a laptop and a computer monitor in the background. |
| 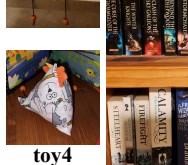 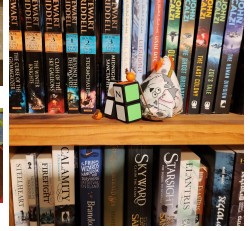 **toy4** | LLaVA: The image features a bookshelf filled with various books. On the shelf, there are two Rubik's cubes, one green and one white, sitting next to each other. The bookshelf is well-stocked with books, occupying most of the shelf space . . .

**RAP-LLaVA**(Ours): The image shows a bookshelf with books on it. There are two toys on the shelf. One is a ⟨toy4⟩, which looks like a cat with orange fuzzy ears and a brown cord attached. The other is a Rubik's cube. |
| 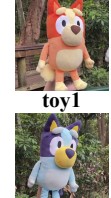 **toy1** 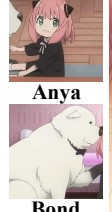 **toy2** 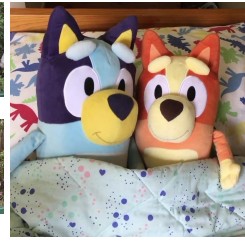 | LLaVA: The image features a bed with two stuffed animals, a dog and a cat, lying on it. The dog is positioned on the left side of the bed, while the cat is on the right side. The stuffed animals are placed close to each other, giving the impression of a cozy and . . .

**RAP-LLaVA**(Ours): The image shows two plush toys, ⟨toy1⟩ and ⟨toy2⟩, lying in bed together. ⟨toy2⟩ is a light blue plush with a purple patch on its head and yellow ears. ⟨toy1⟩ is an orange plush with a yellow belly and a brown nose. They are both looking at the camera. |
| **Anya** 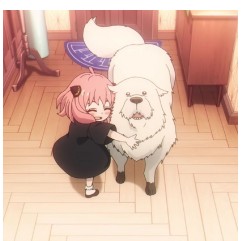 **Bond** | LLaVA: The image features a young girl hugging a large white dog in a room. The girl is wearing a black dress and appears to be enjoying her time with the dog. The dog is positioned close to the girl, and they seem to be sharing a tender moment together . . .

**RAP-LLaVA**(Ours): ⟨Anya⟩ is a young girl with pink hair and big green eyes. She is wearing a black dress and black shoes. She is standing in a room with wooden floors and is hugging a white dog named ⟨Bond⟩. ⟨Bond⟩ is wearing a black bow tie. The dog is standing on its hind legs and is looking at the camera. The room is decorated with a blue rug and a wooden cabinet . . . |

Table 16: Qualitative results of personalized concept recognition obtained by RAP-LLaVA. We use green rectangle to show the bounding box in the image.

| Image | Conversation |
|---|---|
| 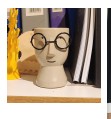 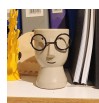 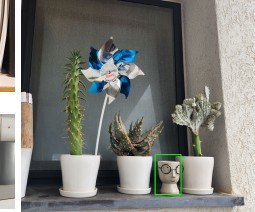 **head*** | Question: Is ⟨head*⟩ in this image? **RAP-LLaVA**: Yes, ⟨head*⟩ is in the photo. Question: Give ⟨head*⟩'s bounding box in the image. **RAP-LLaVA**: [0.57, 0.67, 0.68, 0.84] |
| 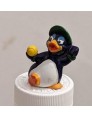 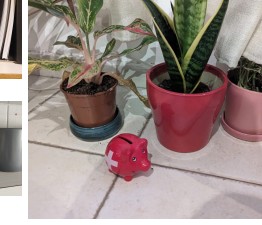 **head*** | Question: Is ⟨head*⟩ in this image? **RAP-LLaVA**: No, ⟨head*⟩ is not part of this photograph. |
| 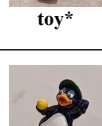 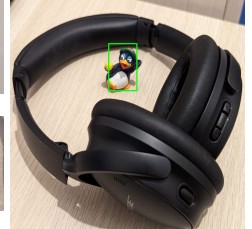 **toy*** | Question: Can you identify if ⟨toy*⟩ is captured in this picture? **RAP-LLaVA(Ours)**: Yes, ⟨toy*⟩ is visible in this picture. Question: Please provide ⟨toy*⟩'s bounding coordinates in the image. **RAP-LLaVA**: [0.42, 0.19, 0.54, 0.39] |
| 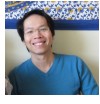 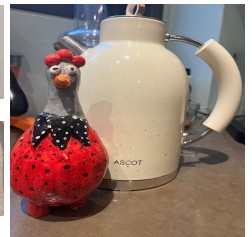 **toy*** | Question: Can you identify if ⟨toy*⟩ is captured in this picture? **RAP-LLaVA(Ours)**: ⟨toy*⟩ is not captured in this image. |
| 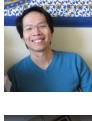 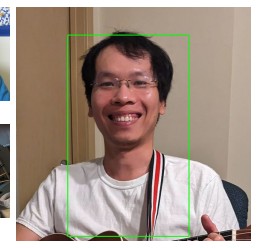 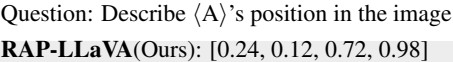 **A** | Question: Can you confirm if ⟨A⟩ appears in this photo? **RAP-LLaVA(Ours)**: Yes, ⟨A⟩ is in this photo. Question: Describe ⟨A⟩'s position in the image. **RAP-LLaVA(Ours)**: [0.24, 0.12, 0.72, 0.98] |
| 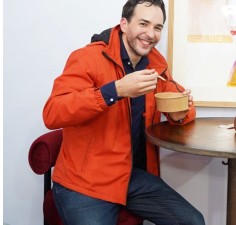 **A** | Question: Can you confirm if ⟨A⟩ appears in this photo? **RAP-LLaVA(Ours)**: ⟨A⟩ is not present in this photograph. |

Table 17: Instructions for visual grounding and recognition.

Visual recognition:
- Is ⟨V⟩ visible in this picture?
- Is ⟨V⟩ in this image?
- Do you see ⟨V⟩ in the photo?
- Is ⟨V⟩ present in this photograph?
- Can you identify if ⟨V⟩ is captured in this picture?
- Is ⟨V⟩ depicted in this image?
- Does the picture feature ⟨V⟩?
- Can you confirm if ⟨V⟩ appears in this photo?
- Is ⟨V⟩ included in this shot?
- Is ⟨V⟩ shown in this image?
- Can you tell if ⟨V⟩ is part of this photograph?
- Is there any sign of ⟨V⟩ in this picture?
- Can you detect ⟨V⟩ in the photo?
- Is ⟨V⟩ captured in this image?
- Do you recognize ⟨V⟩ in this picture?

Visual grounding:
- Give ⟨V⟩'s bounding box in the image.
- Describe ⟨V⟩'s position in the image.
- Please provide the coordinates of the bounding box for ⟨V⟩ in the given image.
- Specify the rectangular boundaries of ⟨V⟩ in the image.
- Give ⟨V⟩'s position in the following image.
- Please provide ⟨V⟩'s bounding coordinates in the image.
- Indicate the bounding box for ⟨V⟩ in the image.
- Show the bounding box for ⟨V⟩ in the picture.
- Specify ⟨V⟩'s bounding box in the photograph.
- Mark ⟨V⟩'s bounding box within the image.

Table 18: Instructions for image captioning.

Image caption:
- Give a caption of the image.
- Give a personalized caption of this image.
- Provide a brief caption of the image.
- Summarize the visual content of the image.
- Create a short caption of the image.
- Offer a short and clear interpretation of the image.
- Describe the image concisely.
- Render a concise summary of the photo.
- Provide a caption of the given image.
- Can you provide a personalized caption of this photo?
- Could you describe this image concisely?

Table 19: Instructions for image description.

Image description:
- Describe the image.
- Give a description of the image.
- Give a description of the image in detail.
- Give a short description of the image.
- Describe the image in detail.
- Please provide a description of the image.
- Can you give me details about the image?
- Could you explain what's shown in the image?

Table 20: Seed questions used for question answering synthesis.

Person:
- What is $\langle H \rangle$'s hair color?
- What is $\langle H \rangle$'s height (estimated)?
- What is $\langle H \rangle$'s skin tone?
- What is $\langle H \rangle$'s eye color?
- What style of clothing is $\langle H \rangle$ wearing?
- Does $\langle H \rangle$ have any visible tattoos?
- Does $\langle H \rangle$ wear glasses or contact lenses?
- Does $\langle H \rangle$ have any facial hair?
- What is $\langle H \rangle$'s approximate age?
- What is $\langle H \rangle$'s build or body type?
- What is $\langle H \rangle$ doing?

Object:
- What color is $\langle O \rangle$?
- What pattern is on $\langle O \rangle$?
- What shape does $\langle O \rangle$ have?
- What size is $\langle O \rangle$?
- What is the texture of $\langle O \rangle$?
- Is $\langle O \rangle$ shiny or matte?
- What material is $\langle O \rangle$ made of?
- Does $\langle O \rangle$ have any patterns or designs on it?
- Is $\langle O \rangle$ new or worn?
- Does $\langle O \rangle$ have any visible brand or logo?
- Is $\langle O \rangle$ functional or decorative?

Multi-concept question:
- What do $\langle C_1 \rangle$ and $\langle C_2 \rangle$ have in common?
- What activity are $\langle C_1 \rangle$ and $\langle C_2 \rangle$ engaged in?
- Where could $\langle C_1 \rangle$ and $\langle C_2 \rangle$ be located?
- What is the most noticeable difference between $\langle C_1 \rangle$ and $\langle C_2 \rangle$?
- What are they doing?

Table 21: Examples of our database. A concept should be provided with an image and its personalized description.

| Image | Concept | Information |
|---|---|---|
|  | Anya | A young girl with pink hair and big green eyes. |
|  | doll* | This is a cute figurine of a girl wearing a pink and blue dress, holding a white bubble. |
|  | toy1 | A plush toy. It is orange with a yellow belly and a brown nose. |
|  | toy2 | This is a plush toy of the bluey character. It is a light blue color with a purple patch on its head, and its ears are yellow. |
|  | statue* | This is a figurine of a cat. The cat has a blue body with yellow, red, and green stripes and a long tail that is also striped. |
|  | cat* | A small ginger kitten with bright blue eyes looks up at the camera. |
|  | H | A young man is wearing a plain tan t-shirt. His hair is short and curly. |
|  | dog* | A white and gray dog with long fur. He has black eyes. |
|  | T | A young woman with blonde hair is wearing a white tank top and blue jeans. |

