# OpenReview forum: "Remember, Retrieve and Generate: Understanding Infinite Visual Concepts as Your Personalized Assistant"
_ICLR.cc/2025/Conference — ICLR 2025 Conference Withdrawn Submission_

### Official Review · Reviewer_MKjC · 2024-11-02

**Soundness:** 4
**Presentation:** 3
**Contribution:** 4
**Rating:** 8
**Confidence:** 4

**Summary:**

This paper introduces a framework called RAP, based on the “remember-retrieve-generate” pipeline, for personalizing MLLMs. This framework allows MLLMs to understand unlimited user-specific concepts, generate personalized captions or respond to user-related questions. Additional, this paper creates a large-scale dataset for personalized MLLM training, with the scale of 260K, including images of specific concepts from different angles, descriptions, visual grounding, image captioning, descriptions, and Q&A instructions and answers. With this dataset, the model learns to generate information based on the provided concept images and their descriptions. Experimental results show the RAP framework’s effectiveness and general applicability.

**Strengths:**

1. Personalized understanding and generation is very important, and this paper presents a recognition-retrieval-conditional generation method that effectively addresses the challenge of scalable personalized memory.
2. During inference, the method is training-free, meaning it doesn’t require additional training as the user’s database expands, making it adaptable to diverse users and capable of real-time updates with new concepts.
3. The paper includes extensive experiments covering various aspects, such as personalized image captioning, VQA, time cost, effects of concept numbers, retriever effects, and more.

**Weaknesses:**

1. In the method, some symbols and calculation explanations are unclear.
2. The training and test data settings in the experiments are not clearly described.

**Questions:**

1. Unclear Symbol Usage or Lack of Explanation

    1). “Q = (I, T)” is defined, but later in lines 242-243, “visual feature Q^I=\Epsilon(I_u^i) is introduced as an n-dimensional vector, reusing “Q” with a different meaning.

    2). Lines 244-245: The notation “Top-K image-text pairs {(I_1, T_1),(I_2, T_2), …(I_k, T_k)}” should clarify what (I_i, T_i) represents and distinguish it from the I, T in the Remember section. As it is, readers need to jump to line 249 to understand, which disrupts flow.

    3). Lines 248-255: It’s unclear how (I_i, T_i) contributes to the Generate section. The text mentions that I_j becomes a visual token Z_j through a pre-trained vision encoder, and then H_j^v through a projector, while T_j becomes H_j^q. But the text doesn’t explain how these tokens are used in generation. Are the tokens simply concatenated? Is M_j represents concat(H_j^v, H_j^p) ?

2. Experimental Settings

    1). In Section 4.1 on personalized image captioning, it’s unclear how much data was used for training vs. testing. This paper only mentions that each concept has 1 training image (150 negative images for MyVLM) and 8-13 test images.

    2). In Section 4.2 on personalized question answering, the testing dataset and amounts of training vs. testing data are also not specified. Even a citation referring to the testing dataset in another paper would be helpful.

3. More Explanation.

    1).  In Table 6, on the two benchmarks MMMU and InfoSeek, why did RAP-LLAva still improve results without using the external KB?

4. [Optional] Suggestion:

    1).  Instead of “allowing models to be trained just once,” maybe use “training-free” to better highlight the model’s advantage, as “trained once” initially sounds like a single round of fine-tuning is required.

    2).  It would be great if your work could further explore the "average number of concepts per image" in addition to Figure 4. Figure 4 focuses on the effects of retrieval, while the "average number of learned concepts per image" would focus more on recognition.

    3).  Will you consider open-sourcing the dataset in the future? This dataset would be highly valuable to the community.

---

### Official Review · Reviewer_7LLv · 2024-11-03

**Soundness:** 1
**Presentation:** 2
**Contribution:** 2
**Rating:** 3
**Confidence:** 3

**Summary:**

This paper introduces RAP (Retrieval Augmented Personalization), a novel framework designed to transform Multimodal Large Language Models (MLLMs) into personalized assistants while preserving their general capabilities. The framework operates through a three-step process: Remember, Retrieve, and Generate. This paper suggest dataset for training personalized MLLMs, which includes data for visual grounding, recognition, image captioning, and question answering.

**Strengths:**

1. This paper created and provided a large-scale, diverse dataset for personalized training of MLLMs.

2. This paper was well written and easy to read.

**Weaknesses:**

1. This paper lacks comprehensive evaluation metrics for the two main experimental tasks. For both personalized image captioning and personalized VQA tasks, the metrics are mainly composed of recall and precision, which focus primarily on the model's ability to detect and identify concepts within the database. The paper omits reporting of traditional automatic metrics for image captioning (e.g.BLEU, METEOR) and standard accuracy metrics for question answering.

2. This paper's main flow appear very similar to its prior work, (e.g.  MyVLM). While the authors emphasize their differentiation through the ability to update the database without retraining, this distinction appears to be primarily achieved through the use of external modules rather than novel technical contributions. Specifically, the claimed "training-free" advantage seems to be inherited from using YOLO-world, an existing open-vocabulary object detection model that is inherently training-free. Additionally, the performance improvements appear to be largely attributed to the use of FAISS, a powerful existing model for face similarity calculations.Therefore, the paper's claimed distinctions from previous work appear to lack technical novelty. Furthermore, the topic itself cannot be considered particularly novel, as there are several previous works addressing similar problems in personalization of multimodal models.

**Questions:**

1. The paper mentions that due to the context length limitations of LLaVA and Phi-V models, only 2-3 concepts can be retrieved.Are these 2-3 augmented concepts different possible target concepts When multiple concepts are retrieved, how can we determine which one is the target concept? If there is no way to identify which is the target concept, why do we need to retrieve multiple concepts instead of just one?

2. It would be good to add traditional metrics that can measure both concept identification and task performance simultaneously in personalized image captioning and question answering. While the authors demonstrate maintained performance on datasets like MMMU and InfoSeek, they should also provide metrics that evaluate the intrinsic quality of image captioning and question answering performance during personalized tasks.

---

### Official Review · Reviewer_LU4Z · 2024-11-04

**Soundness:** 3
**Presentation:** 2
**Contribution:** 1
**Rating:** 5
**Confidence:** 4

**Summary:**

This paper proposed a retrieval augmented generation method to enable personalized visual understanding. To enable personalized user information retrieval, the paper proposed to extract visual objects for retrieval. In addition, retrieved user information will be added to MLLMs' prompt.

**Strengths:**

1. The personalization understanding problem in MLLMs is crucial.

2. A very simple but effective RAG method is proposed.

**Weaknesses:**

1. The authors could summarize the extracted concepts and some statistics about sample distributions. It would be better to understand the challenge of the proposed task.

2. Although the claimed personalization sounds insightful and useful, most of the concepts demonstrated in the paper (e.g., cat, dog, doll, toy, etc.) are rather general visual objects which might also be recognized correctly by general MLLMs without RAG.

3. Since much of the provided information in the personalized database (e.g., descriptions about toys, dolls, cat, dog, etc.) is descriptive texts, it would be arguable that if the performance improvement comes from personalization or more accurate visual ground-truth.

**Questions:**

1. Could the authors provide some more insightful examples which could demonstrate the necessity of including personalized information in VQA? Specifically, would the personalized information influence the answer to some compositional reasoning tasks and change the answers?

---

### Note · Authors · 2024-11-15

**Comment:**

Thanks for constructive comments from the reviewers. After thorough deliberation, we have decided to withdraw our submission. We sincerely thank you for your time and feadback.

**Withdrawal Confirmation:**

I have read and agree with the venue's withdrawal policy on behalf of myself and my co-authors.